# Sol-Gel Synthesis of Caffeic Acid Entrapped in Silica/Polyethylene Glycol Based Organic-Inorganic Hybrids: Drug Delivery and Biological Properties

Luigi Vertuccio [1], Liberata Guadagno [2], Antonio D'Angelo [1,3], Veronica Viola [2], Marialuigia Raimondo [2] and Michelina Catauro [1,*]

1    Department of Engineering, University of Campania "Luigi Vanvitelli", Via Roma 29, 81031 Aversa, Italy
2    Department of Industrial Engineering, University of Salerno, Via Giovanni Paolo II, 132, 84084 Fisciano, Italy
3    Department Environmental, Biological and Pharmaceutical Sciences and Technologies, University of Campania "Luigi Vanvitelli", Via Vivaldi 43, 81100 Caserta, Italy
*    Correspondence: michelina.catauro@unicampania.it

**Abstract:** The failure of medical devices, such as bones prosthesis, is mainly due to inflammatory and infectious phenomena. Entrapping anti-inflammatory and antimicrobial agents inside the biomaterial matrix could avoid these phenomena. In this context, inorganic/organic silica (S)/polyethylene glycol (P)/caffeic acid (A) hybrid systems were synthesized via the sol-gel method with different weight percentages of P and A. Fourier-transform infrared (FT-IR) revealed that caffeic acid undergoes an oxidizing phenomenon in the sol-gel synthesis condition. Additionally, the formation of a hydroxyapatite layer on hybrid surfaces was demonstrated by employing the Kokubo test and analyzing the samples using scanning electron microscopy, X-ray diffraction, and FT-IR. Moreover, further characterization of the antimicrobial activity of the synthesized biomaterials was carried out using the Kirby–Bauer test. Finally, UV-Vis measurement was useful to evaluate the caffeic acid kinetic release in simulated body fluid (SBF) at 37 °C. The kinetic study disclosed that the hybrid materials without polyethylene glycol had faster release rates than the ones obtained without the organic polymer.

**Keywords:** sol-gel; hybrid materials; drug delivery; bioactivity

## 1. Introduction

Biomaterials (BMs) are natural or synthetic substances that interact with the human body, enhancing the repair or replacement of tissues because of their biocompatibility [1]. BMs can be divided into different classes: metallic, ceramic, glassy, polymeric, composite, and biodegradable polymers [2,3]. Among these classes, bioceramics and bioglasses have attracted great interest from researchers because of their ability to promote osteointegration [4–7]. The introduction of new implants to a living body may cause inflammation phenomena with consequent infection processes. These phenomena could be avoided by producing hybrid composite BMs with anti-inflammatory and antimicrobial agents, improving the therapeutic effect, and enhancing controlled drug delivery [8].

Traditional bioceramic and bioglass materials used for the implant are made by high-temperature processes that may be up to 1000 °C, and they do not allow the entrapping of thermolabile compounds, such as drugs. To avoid this drawback, the use of sol-gel synthesis allows the incorporation of a bioactive agent (anti-inflammatory or antimicrobial drug) [9–11].

The sol-gel process is a colloidal route used to synthesize any material with an intermediate stage, including a sol and/or a gel state, starting from a molecular precursor (e.g., metal salts, alkoxides, organic monomers, oligomers) or colloidal particles (e.g., graphene oxide sheet, carbon nanotubes) [12].

In the case of the sol-gel synthesis for organic/inorganic hybrid glassy biomaterial, it occurs with hydrolysis and polycondensation of a molecular precursor, generally, a metal alkoxide $M(OR)_X$ where M is a metal (such as Al, B, Si, Ti, Zr, etc.) and R is an alkyl group, by heating and stirring processes [13]. These reaction steps led to the formation of a gel structure, which turns into glass material after the drying process [14,15]. Despite some disadvantages (costs of the precursor, long gelation time, and strong reduction in the gel volume after the drying procedure [12,14]), the sol-gel allows the control of reaction parameters, such as the concentration of reactants, temperature, and catalysts; therefore, the control of materials synthesized, such as the morphology and mechanical properties. Current state-of-the-art research reflects how sol-gel chemistry shows several advantages since it allows the possibility to synthesize functional materials, which are usable in various fields [16–19].

Recent studies have been conducted on the feasibility of synthesizing via the sol-gel route hybrid systems (formed by silica, polyethylene glycol, and quercetin) having antibacterial properties [20,21]. Like quercetin, caffeic acid (A), is one of the major nutritional antioxidants and belongs to the family of flavonoids. It is ubiquitous in vegetables, fruits, tea, wine, and food supplements in the form of extracts and/or pure forms [22]. Caffeic acid, also known as 3,4-dihydroxycinnamic acid, is a phenolic acid derived from hydroxycinnamic acid. This organic acid has many effects on various diseases, such as osteoporosis, certain cancers, lung diseases, cardiovascular diseases, and ageing [23,24]. Moreover, it also has interesting biological properties, such as antimicrobial, fungicides, and antioxidants [25,26]. It has been reported that caffeic acid shows antimicrobial potential against *Staphylococcus aureus*, *Staphylococcus epidermidis*, *Klebsiella pneumoniae*, *Serratia marcescens*, *Proteus mirabilis*, *Escherichia coli*, *Pseudomonas aeruginosa*, *Bacillus cereus*, *Micrococcus luteus*, *Listeria monocytogenes*, and *Candida albicans* strains [26–29]. The antimicrobial activities of caffeic acid are partly due to its lipophilic nature that allows the interaction with the cytoplasmic membrane of the microbial cells, especially with the hydrophobic portion [27].

Many researchers found that the use of polyethylene glycol (P) as an organic additive in the sol-gel methods allows for protecting and altering the release rate of the drug embedded in the matrix [30–32]. In addition, the presence of P improves cell adhesion and growth by increasing the hydrophilicity of the materials and the controlled release of therapeutic molecules. Indeed, it has been demonstrated that the variation of the P content of polylactic acid/polyethylene glycol copolymers allows for controlling the adsorption of adhesion proteins and cell adhesion, due to the occurrence that cell adhesion takes place only in the presence of serum proteins [33].

In our previous paper, we demonstrated the feasibility to synthesize via the sol-gel route with hybrid organic/inorganic materials made up of silica and different weight percentages (wt%) of caffeic acid (from 0 to 20 wt%) [34]. Additionally, it has been demonstrated that the occurrence of some structural modifications, which nevertheless ensured a strong radical scavenging capacity of samples with 15 and 20 wt% of caffeic acid incorporated into the systems. In this paper, the release study of the entrapped drug was also investigated. Moreover, in such systems polyethylene glycol, another organic compound had been added to shed light on the antimicrobial effect and the influence on the release of the drug.

Here, hybrid organic-inorganic materials were synthesized via acid-catalyzed sol-gel reactions, where the $SiO_2$ (labelled S) is the inorganic phase, whereas the organic precursors are the polyethylene glycol (labelled P) and the caffeic acid (labelled A). The SPxAy hybrids were composed of 0, 6, and 12 wt% (x) of P and 5, 10, 15, and 20 wt% (y) of A. The formation of new chemical bonds or weak interactions (e.g., van der Waals forces or hydrogen bonds) among the organic and inorganic phases inside the hybrid materials was evaluated using Fourier transform-infrared (FT-IR) spectroscopy. The bioactivity, as the ability to facilitate the hydroxyapatite layer formation on the hybrid materials, was analyzed using scanning electron microscopy/energy-dispersive X-ray spectroscopy (SEM/EDS), FT-IR and X-ray diffraction (XRD) after the materials were soaked at 37 °C in SBF solution (Simulated

Body Fluid) for 21 days. The in vitro release study of caffeic acid was evaluated using UV-Vis spectroscopy, whereas the antibacterial properties were assessed against both Gram-(*Escherichia coli*, *Pseudomonas aeruginosa*) and Gram+ *(Staphylococcus aureus, Enterococcus faecalis*) microbial strains.

## 2. Materials and Methods

### 2.1. Sol-Gel Synthesis of the Materials

The SPxAy hybrid materials with different percentages of Polyethylene Glycol (P, MW = 400, Sigma–Aldrich, St. Louis, MO, USA) and Caffeic Acid (A, $C_9H_8O_4$, MW = 180.16, Sigma–Aldrich, Milan, Italy) were synthesized using the sol-gel route (starting from our previous paper [34]) as outlined in Figure 1.

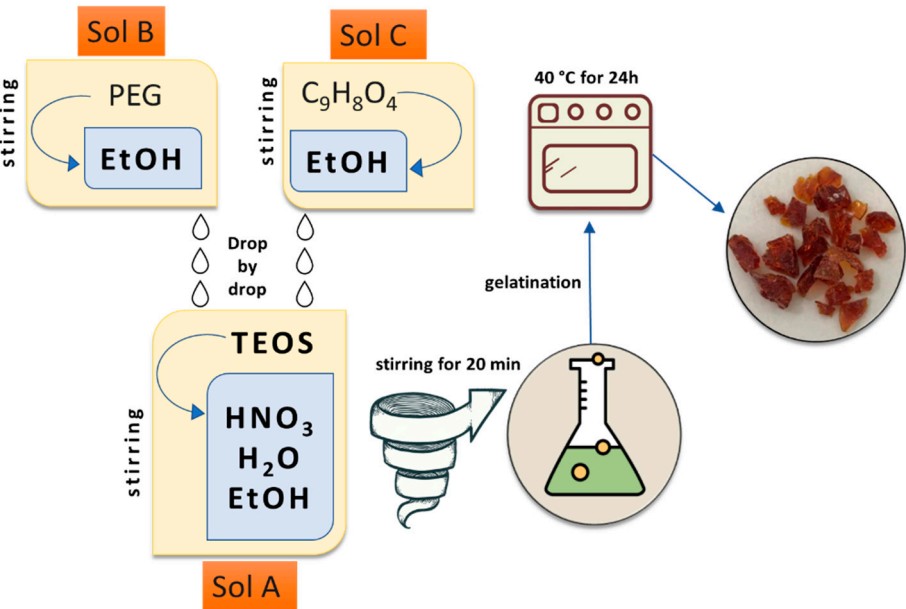

**Figure 1.** Flowchart of the used sol-gel procedure.

Briefly, tetraethyl orthosilicate (TEOS, $Si(OC_2H_5)_4$, Sigma–Aldrich, St. Louis, MO, USA) was added to a solution of distilled water and ethanol 99% (EtOH, Sigma–Aldrich, St. Louis, MO, USA) under magnetic stirring, forming Sol A, while polyethylene glycol and caffeic acid were dissolved in EtOH (Sol B and C). After the formation of Sol A, Sol B and C were added to Sol A drop-by-drop under continuous stirring at T = 30 °C. Within the temperature synthesis, which influences gelation occurrence, nitric acid ($HNO_3 \geq 65\%$, Sigma–Aldrich, Milan, Italy) was also added to the final solution to increase the hydrolysis rate reactions [13,35]. The molar ratios used for the reagents were $TEOS:HNO_3:EtOH:H_2O = 1:1.7:6:2$. After the gelation process, the wet gels were put in an oven at 40 °C for 24 h. All the hybrid materials were synthesized following the reported procedure and were labelled with the following name code: SPxAy, where x represents the weight percentages of P compared to the S weight amount, whilst y represents the weight percentages of A to S content (e.g., the hybrid material with P = 0 wt% and caffeic acid = 5 wt% is SA5, while SP6A5 is the hybrid material with polyethylene glycol = 6 wt% and caffeic acid = 5 wt%). Table 1 summarizes all the synthesized hybrids with their code names.

**Table 1.** Compositions of synthesized hybrids with their acronyms. S = SiO$_2$; Ay= caffeic acid wt%; Px = polyethylene glycol wt%.

| Sample | Polyethylene Glycol (wt% in Relation to SiO$_2$ Matrix) | Caffeic Acid (wt% in Relation to SiO$_2$ Matrix) |
|---|---|---|
| SA5 | 0 | 5 |
| SA10 | 0 | 10 |
| SA15 | 0 | 15 |
| SA20 | 0 | 20 |
| SP6A5 | 6 | 5 |
| SP6A10 | 6 | 10 |
| SP6A15 | 6 | 15 |
| SP6A20 | 6 | 20 |
| SP12A5 | 12 | 5 |
| SP12A10 | 12 | 10 |
| SP12A15 | 12 | 15 |
| SP12A20 | 12 | 20 |

### 2.2. FT-IR Analysis

The FT-IR technique was used to investigate the interactions between the components. Transmittance spectra were obtained by using a Prestige 21 Shimadzu (Japan) instrument. The spectra were recorded in the 400–4000 cm$^{-1}$ region, with a resolution of 2 cm$^{-1}$ (64 scans). FT-IR was conducted on KBr disks containing 2 mg of sample powder and 198 mg of salt. The recorded spectra were processed using IRsolution and Origin software.

### 2.3. Bioactivity Test

The bioactivity test was carried out by the hydroxyapatite (HA) forming ability of the synthesized hybrid materials as reported in [36,37]. The HA formation was evaluated after soaking 250 mg of sample disks into 50 mL of simulated body fluid (SBF), which is a solution whose ion concentrations are nearly equal to those of human blood plasma [36], at 37 °C for 21 days. Ion species depletion was avoided using the replacement of the SBF solution each 48 h. After 21 days, the sample disks were gently washed and dried in a glass desiccator (Sigma–Aldrich, St. Louis, MO, USA), and then subjected to FT-IR, X-ray diffraction (XRD), and scanning electron microscopy with energy-dispersive X-ray spectroscopy (SEM/EDS) analyses. FT-IR was carried out as described in the previous paragraph, whilst XRD analysis was performed with a Philips 139 diffractometer equipped with a PW 1830 generator, a tungsten lamp, and a Cu anode. Lastly, SEM images were acquired with Phenoma XL G2 (Alfatest, Italy), whereas EDS information was obtained by using Live EDS tool from the same machine.

### 2.4. Antimicrobial Activity

The antimicrobial activity of synthesized hybrid materials was performed according to Kirby–Bauer Protocol [20,38]. To this aim, the hybrid materials were ground, obtaining powders, which were further compressed into 100 mg disks. All the sample disks were sterilized using UV light for 1 h to avoid external contamination during the tests. Antimicrobial properties were assessed against both Gram+ bacteria, such as *S. aureus* (ATCC 25923) and *E. faecalis* (ATCC 29212), and Gram- bacteria, such as *E. coli* (ATCC 25922) and *P. aeruginosa* (ATCC 27853). These bacteria were chosen as they are known as bacteria that cause nosocomial infections during patient hospitalization [39]. All the pelletized bacterial strains were diluted in distilled saline water (0.9% NaCl), achieving suspensions of 10$^9$ CFU/mL, that were plated onto the respective agar-based media (whose preparation was reported elsewhere in [20]). After the bacterial plating, all the sterilized samples were left in the middle of the plates and then they were placed in the respective incubators that mimic the growth condition of each bacterium. Indeed, *E. coli* was incubated at 44 °C for 24 h, *P. aeruginosa* and *E. faecalis* were incubated at 36 °C for 48 h, whereas *S. aureus* was

incubated at 36 °C for 24 h. After the incubation times, all the bacterial plates were removed from the incubators and the inhibition halo diameters (IHDs) were measured.

### 2.5. In Vitro Release

The in vitro release study was performed to evaluate both the capability of the synthesized materials to release the entrapped drug and the kinetics. The test was performed with a Shimadzu UV mini-1240 and the measurements were taken at a wavelength of λ = 216.0 nm, corresponding to the maximum absorption of oxidized caffeic acid. Indeed, the absorption is ascribed to the C=O group about the electronic transition π–>π* [34]. This choice is based on the fact that the acid catalyst causes the aromatic destruction system with shifts of maximum adsorption wavenumber from 324.0 nm to 216.0 nm [34]. Measurements of oxidized caffeic acid releases were carried out in samples soaked in 10 mL of SBF.

The calibration curve (Figure 2) was made by taking the absorbance measurement of standards prepared in SBF and establishing the relation with the solution concentrations and absorbance at the length of 216.0 nm, with $R^2$ = 0.9993. The lower limit of the calibration curve was 0.25 mg/L, whereas the higher limit was 20.0 mg/L.

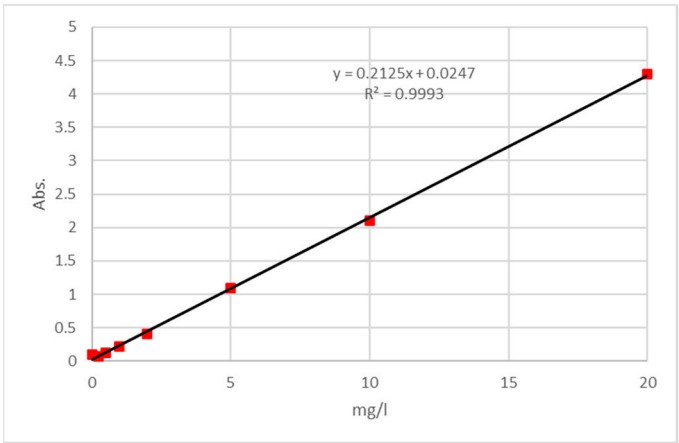

**Figure 2.** Calibration curve of Caffeic acid.

## 3. Results and Discussion

### 3.1. FT-IR Analysis

Recently, it has been demonstrated that in the hybrid silica/polyethylene glycol/quercetin biomaterials, obtained using the acid-catalyzed sol-gel route, the phenolic drug compound undergoes an oxidizing phenomenon, which led to the formation of a hybrid that still stores the antioxidant and antimicrobial activity [20]. Starting from this finding and the previous results based on silica and caffeic acid hybrid materials, the occurrence of chemical modification was investigated using the comparison of the FT-IR spectra of pure caffeic acid (Figure 3A) and the oxidized one (Figure 3B) obtained by using the sol-gel procedure without adding TEOS and P inside the solutions, followed by the solvent removal. In the pure caffeic acid spectrum, Bands of about 3424 and 3233 cm$^{-1}$ are attributed to the -OH stretching vibration. Bands at 2982 and 2912 cm$^{-1}$ are ascribed to the C-H vibration modes. The intense band at 1645 cm$^{-1}$ is assigned to the stretching of carbonyl group C=O [40–42]. Moreover, the intense bands at 1625 cm$^{-1}$ and 1450 cm$^{-1}$ are attributed to olefinic C-C stretching modes. Thus, the signal at 1217cm$^{-1}$ is imputed to aromatic bending C-H modes. Frequencies lower than 1120 cm$^{-1}$ might be due to the C-C-C bending modes of the aromatics system except for the signals at 815 and 648 cm$^{-1}$ that are assigned to bending modes of the carbonyl group [34,41,42]. As a confirmation of oxidation occurrence, there are some shifts of the main signals both to lower and higher wavenumbers, followed by the formation of new absorption bands. Indeed, the two sharp absorption peaks in the range of 3500–3200 cm$^{-1}$ (Figure 3A) changed into a main broad peak at 3431 cm$^{-1}$, with a shoulder located at 3250 cm$^{-1}$ (Figure 3B), probably due to the increase in -OH and

H-bonds [43] in the oxidized structure. Moreover, because of the oxidization, the signals at 2982 and 2912 cm$^{-1}$ (Figure 3A), attributed to -CH$_3$ sp3 stretching, shifted at 2987 and 2908 cm$^{-1}$. Furthermore, there was also a shift of the C=O vibration from 1645 cm$^{-1}$ to 1745 cm$^{-1}$. This shift to higher wavenumbers could be explained by the stopping of π electron delocalization of the C=O group [40,41], because of the oxidization phenomenon. This latter band within the bands at 860 cm$^{-1}$, attributed to C-H alkene bending, and the band at 1635 cm$^{-1}$, attributed to C=C alkene stretching, may suggest that the oxidized caffeic acid still contains the active part of its molecule.

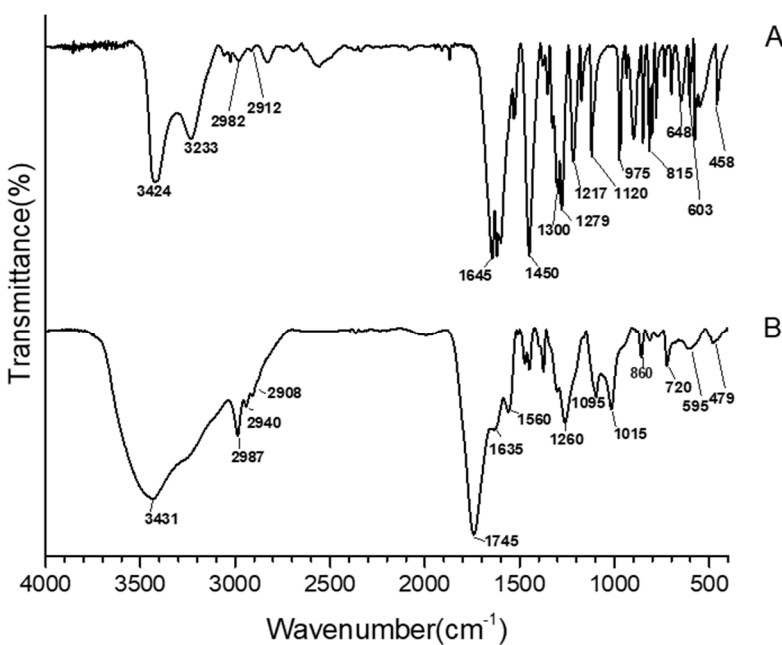

**Figure 3.** FT-IR spectra of caffeic acid (**A**) and oxidized caffeic acid (**B**).

To better understand the chemical influence of the inorganic and organic components of the synthesized hybrid materials, in Figure 4A, the spectra of the silica and the caffeic acid are compared to the SP12Ay systems, whereas in Figure 4B, the spectra of the silica and polyethylene glycol are compared to SPxA20 systems. SiO$_2$ showed the -OH stretching and bending at 3450 cm$^{-1}$ and 1640 cm$^{-1}$. The shoulder at 3700–3500 cm$^{-1}$ could be ascribable to the -OH of the silanol groups [44–46]. The Si-O-Si asymmetric stretching is assigned to the band at 1077 cm$^{-1}$ with the shoulder at 1200 cm$^{-1}$ [44,47,48]), whereas the peak absorption at 800 and 460 cm$^{-1}$ are attributed to Si-O-Si and Si-OH bending vibrations [45]. In addition, the band at 1385 cm$^{-1}$ is assigned to residual HNO$_3$ [42]. In all the spectra shown in Figure 4A, at a fixed amount of silica and polyethylene glycol (12 wt%), the increment in the caffeic acid amount corresponds to an increase in the intensity of the band detected at 1745 cm$^{-1}$ (C=O vibration [36]). This is further proof that caffeic acid is present in its oxidized form in all the synthesized systems. Figure 4B pointed out that at a fixed amount of silica and A (20 wt%), an increase in polyethylene glycol corresponds to an increase in the C-H twisting and C-O-C ether stretching [40,49] detected at 1240 and 1100 cm$^{-1}$, which also affect the shape of the shoulder at 1200 cm$^{-1}$ in all the spectra. Moreover, the signals detected at 2900–2800 cm$^{-1}$, assigned the C-H vibrations, increase with the increasing of P content. Additionally, all the bands under the wavenumber range 1000–460 cm$^{-1}$ are almost superimposable to those of pure SiO$_2$ [42]. Finally, The co-presence of P, A, and S absorption peaks suggests the formation of hybrids, in which both the inorganic and organic parts could interact with hydrogen bonds, as also reported in [20].

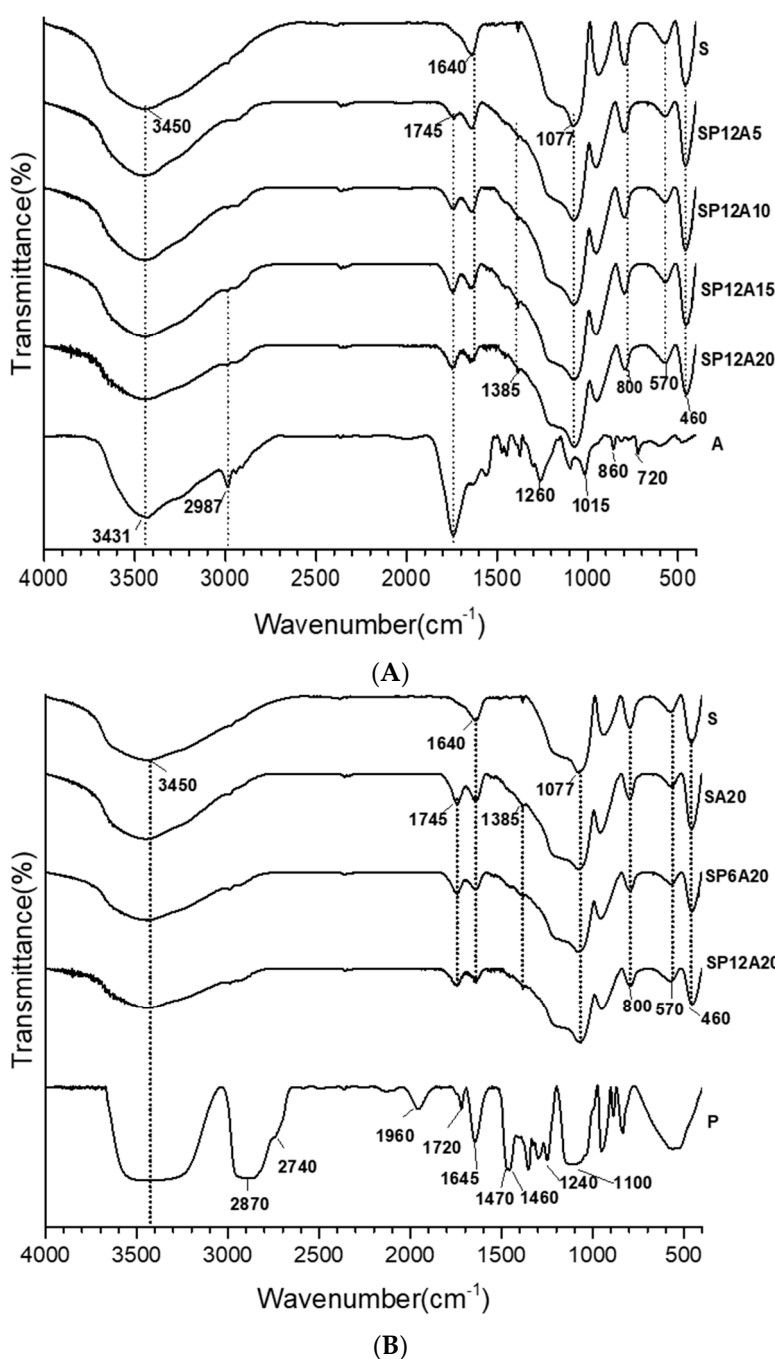

**Figure 4.** FT-IR spectra of (**A**) SP12Ay and (**B**) SPxA20 hybrid systems.

### 3.2. Bioactivity

The HA forming-ability of the synthesized hybrid materials was performed in vitro using the Kokubo Test [37]. To this aim, the samples were immersed in SBF solution for 21 days and the nucleated HA on the surfaces of the materials was investigated using FT-IR, SEM/EDX, and XRD analysis. Figure 5 reports the obtained FT-IR spectra of SP6A20 and SP12A20 hybrid biomaterials after soaking in SBF. The spectra revealed the presence of new peaks at 635, 585, 560, and 470 cm$^{-1}$ that are related to P-O bending and vibration modes [42,50,51].

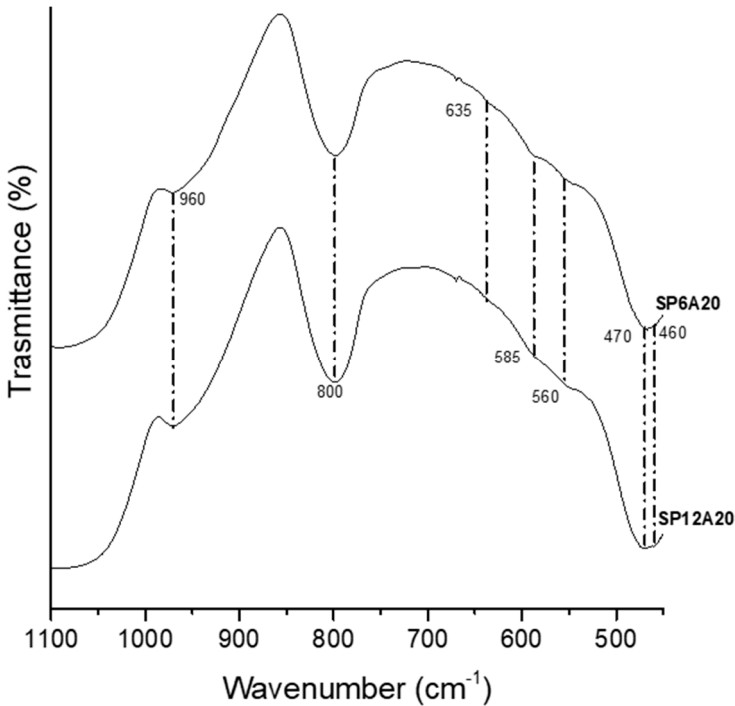

**Figure 5.** FT-IR spectra of washed SP6A20 and SP12A20 after soaking in SBF for 21 days.

Moreover, the SEM/EDS image (Figure 6) of SP12A20 hybrid biomaterial confirmed the formation of the HA. Indeed, the sample surface was almost covered by globular-shaped HA particles [52] whose elemental composition was composed mainly of Ca and P, with an atomic ratio equal to 1.67. This finding is also in accordance with another similar system in which SP systems were doped with indomethacin [53].

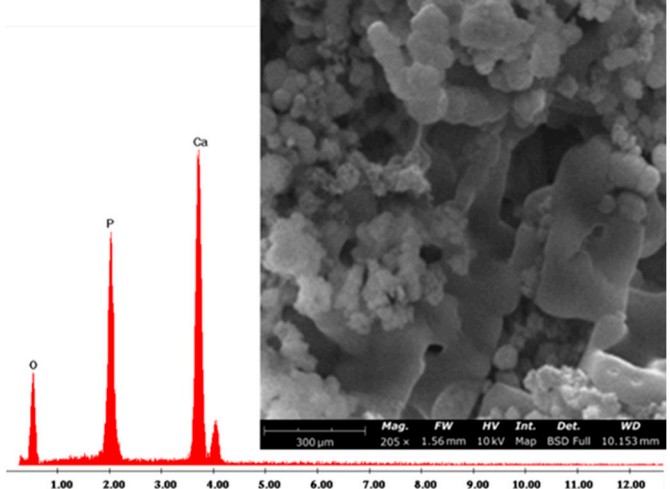

**Figure 6.** Representative SEM/EDX image of SP12A20.

Furthermore, the XRD spectrum, reported in Figure 7, revealed that the spheres nucleated on the hybrid surfaces possess the typical HA crystalline peaks with hkl indices of (002), (211), (300), (202), (310), (002), (222), and (213), obtained by matching the spectrum with one found in ICDD database.

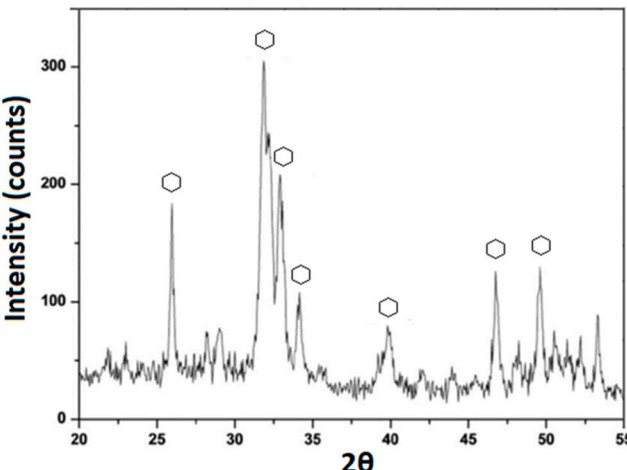

**Figure 7.** XRD spectrum of SP12A20 after 21 days of exposure to SBF.

### 3.3. Antibacterial Activity

The antimicrobial ability of the synthesized hybrid biomaterials was assessed using the Kirby–Bauer protocol. This latter is applied to evaluate the sensitivity of pathogenic aerobic and facultative anaerobic bacteria to several antimicrobial compounds, assisting a physician in selecting treatment options for his or her patients [40]. In this paper, both Gram+ (*S. aureus* and *E. faecalis*) and Gram- (*E. coli* and *P. aeruginosa*) bacteria were tested in the presence and absence of the SPxAy hybrid biomaterials. All these bacterial strains cause nosocomial infections, which are healthcare-associated infections that occur in patients under medical care [54,55].

The antimicrobial activity was expressed by measuring the IHDs as reported in paragraph 2.4. Figure 8 describes, as an example, the acquired images of the bacterial growths of *S. aureus* and *P. aeruginosa* incubated with SA20, SP6A20, and SP12A20 hybrid biomaterials, while Figure 9 reports all the IHDs obtained for each synthesized system.

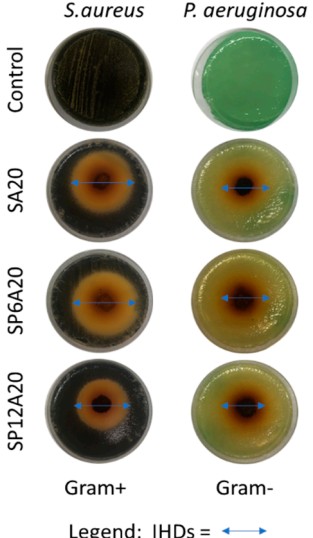

**Figure 8.** Images of the inhibition halos obtained on *S. aureus* and *P. aeruginosa* strains after the incubation with SA20, SP6A20, and SP12A20 hybrid biomaterials.

Caffeic acid and its derivatives are already known for their antibacterial activity [56,57]. Here, it has been proved that they are still active in the synthesized hybrids. Indeed, when bacteria were grown in the presence of the SAy hybrid biomaterials, there was an increased activity against all the Gram+ and Gram- assayed, especially regarding *E. coli* [58], as

a function of the A content. SA5 showed lower activity in the presence of *P. aeruginosa* (IHD = 2.08 ± 0.07 cm) compared to the IHDs (2.60 to 3.00 cm) obtained with the other bacteria, while SA10, SA15, and SA20 samples showed a slightly increased activity against *S. aureus*, *E. faecalis*, and *P. aeruginosa*, whereas against *E. coli*, the IHDs increased from 3.00 to 4.50 cm as the increase in the caffeic acid amount inside the systems. The increment of the antibacterial activity in these systems can be justified by the fact that the catechol functional group present in caffeic acid can be totally or partially oxidized to form the orthoquinone or the semiquinone radical [56]. These two forms can undergo subsequent oxidative processes that can damage and/or destroy DNA and protein structures [29,59–63].

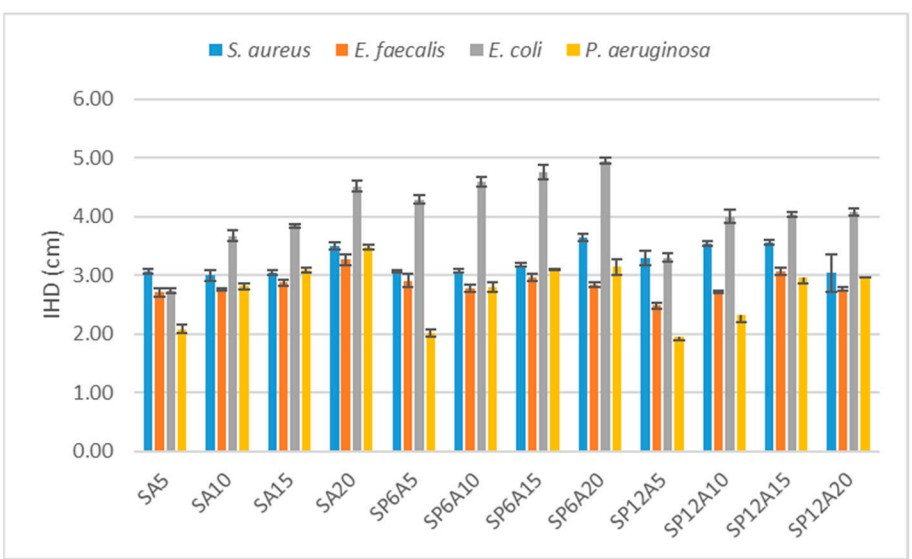

**Figure 9.** Comparison of the inhibition halo diameters for all the synthesized hybrids assayed in the presence of *S. aureus*, *E. faecalis*, *E. coli*, and *P. aeruginosa*.

Comparing the samples with 5, 10, and 15 wt% of caffeic acid only, and the one containing the same amount of caffeic acid but with the addition of 6 wt% of P, it is possible to observe that the addition of P does not affect the antibacterial activity against the Gram+ strains and *P. aeruginosa*. However, as found in the literature regarding the polyethylene glycol [64], the addition of 6 wt% of P seems to highly influence the antibacterial activity against *E.coli* strains, indeed, the IHDs recorded for this strain, of SA5, SA10 and SA15 are 2.74 ± 0.04 cm, 3.67 ± 0.09 cm and 3.84 ± 0.03 cm, respectively, while the IHDs of SP6A5, SP6A10 and SP6A15 are 4.29 ± 0.07 cm, 4.59 ± 0.09 cm, and 4.76 ± 0.12 cm, respectively.

On the other hand, with the same amount of caffeic acid (20%), the addition of 6% of P leads to a slight increase in the inhibition halos for *E. coli* and *S. areus* and a slight decrease for *P. aeruginosa* and *E. faecalis* strains.

The same comparisons made with samples having 12 wt% of P showed that in general, a high amount of P negatively affects the antibacterial activity of all samples. Although in many cases the addition of 12 wt% of P leads to a higher antibacterial activity, the value reached is less high than the one reached with the addition of 6 wt% of P. This finding is also in accordance with the data recorded for hybrid systems based on silica, silica/PEG, and silica/PEG/Quercetin [20]. Indeed, even if silica and silica/PEG systems possess antimicrobial activity, this latter increased with the presence of the drug and 6 wt% of PEG and decreased by increasing the PEG amount [20].

The reason why a higher amount of P leads to a lower antibacterial activity could be attributed to different reasons. Surely P is known to modulate the release of drugs [64], so a slower release of A could lead to a minor presence of free A and so a lower presence of the group responsible for the antibacterial activity. Furthermore, even in the presence of both P and A, it is not possible to state that these have a synergistic action, and on the contrary, they could negatively influence each other.

### 3.4. In Vitro Release

Figure 10A–C illustrates the cumulative kinetics as a percentage of the drug released, whilst Figure 10D–F reports the release rates of the drug over experimental time. The rate of each stage might depend on the formation of hydrogen bonds among the carbonyl group of drugs with the -OH group of silica and aqueous solvent (SBF) [65]. SAy samples showed the fastest release (wt%) and rate (mg/min) (Figure 10A,D). In particular, the SA5 hybrid released all the drug within 1 h, while SA10, SA15, and SA20 released up to 90 wt% of the drug within 6 h, and the release was completed after 12 h for the SA10 hybrid and 24 h for SA15 and SA20. An increase in P amount of 6 wt% (Figure 10B,E) led to a slight decrease in the release rate. Indeed, SP6A5 released 90 wt% of the drug in 6 h and released all the amount after 24 h. An increase in caffeic acid amount (from 10 to 20 wt%) corresponds to a decrease in wt% and rate release, which seems to be more controlled. Furthermore, the data reported in Figure 10C,F, related to SP12Ay hybrid systems underline that an increase in polyethylene glycol and caffeic acid amount led to a drug-release rate lower than the samples without the P, which is also confirmed by the graphs shown in Figure 11. According to [66], which investigated the properties of silica matrix synthesized via sol-gel route by using different TEOS:$H_2O$ ratios, a silica matrix with TEOS:$H_2O$ = 1:2 ratio has a nanometric scale (12.50 nm < x < 32.01 nm), a high surface area (347.31 $m^2$/g < x < 549.92 $m^2$/g), a total pore volume ranging from 0.41 to 0.51 $cm^3$/g, and isotherm hysteresis type H2. These features could explain the reason why there is a faster release of the drug in the matrices without P. Moreover, the presence of P positively affects the release of the drug increasing the control of release over time. Finally, the kinetic study suggests a two-step mechanism: the first fast step occurs by dissolution and diffusion of the caffeic acid entrapped onto the material surface, while the second slow step occurs with the dissolution and diffusion inside the material clusters.

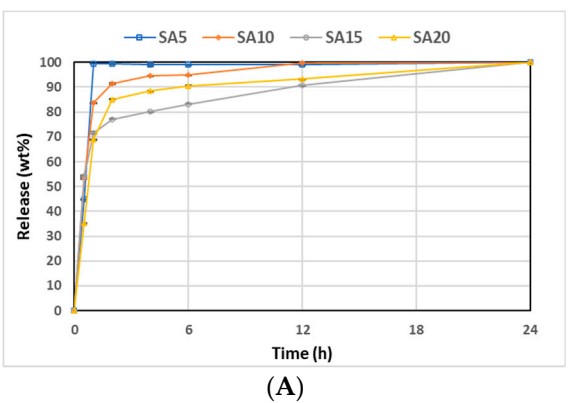

(**A**)

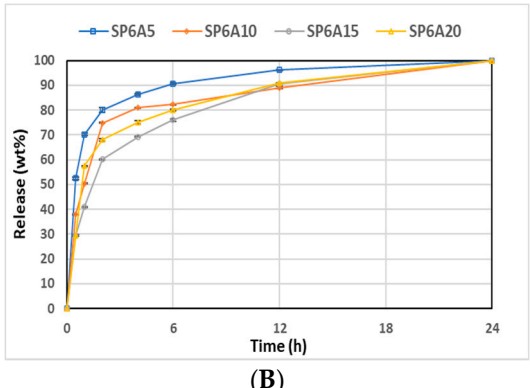

(**B**)

**Figure 10.** *Cont.*

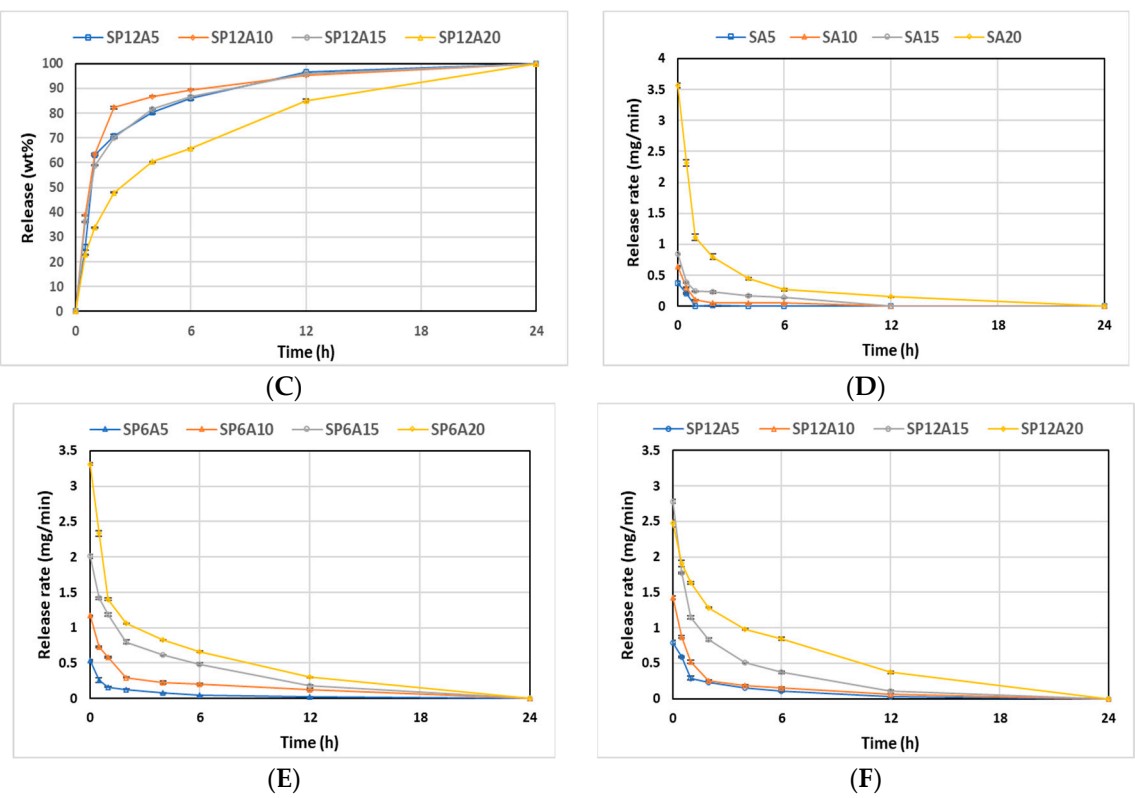

**Figure 10.** (**A**) SAy, (**B**) SP6Ay, and (**C**) SP12Ay drug-release plots. (**D**) SAy, (**E**) SP6Ay, and (**F**) SP12Ay release rate plots of the hybrid biomaterials.

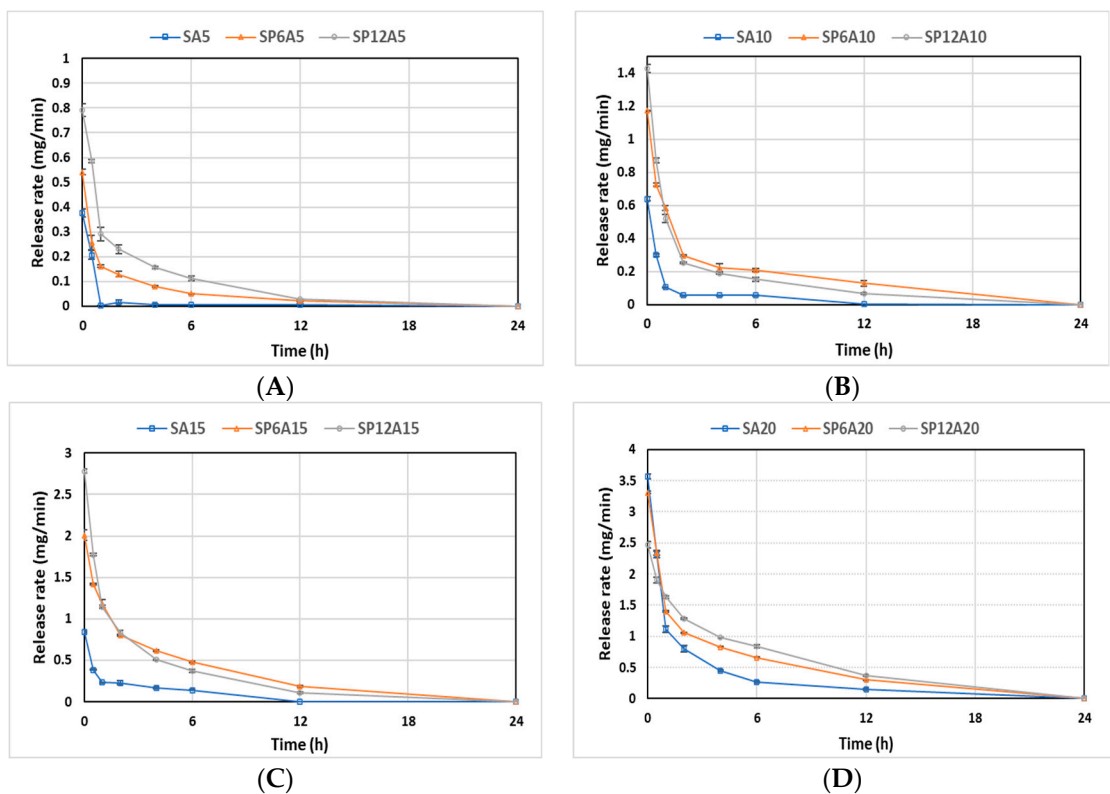

**Figure 11.** Release rate plot of (**A**) SPxA5, (**B**) SPxA10, (**C**) SPxA15, and (**D**) SPxA20 hybrid biomaterials.

## 4. Conclusions

The sol-gel method was successfully applied to obtain hybrid biomaterials made up of silica as an inorganic component and polyethylene glycol and caffeic acid as organic compounds. The latter was selected as a drug for its natural antioxidant properties. FT-IR analysis disclosed that during the sol-gel synthesis, caffeic acid undergoes an oxidizing process and that all the hybrids revealed both the main vibration bands of the organic phases and the inorganic phase. Moreover, it also proved that in all the SPxAy hybrid biomaterials, the absorption bands are from both the inorganic and organic phases. SEM/EDX, XRD, and FT-IR demonstrated that all the samples were involved in the first step of bioactive materials for bone regeneration as the soaking in the SBF solution led to the formation of HA granules, which were found on the hybrid surfaces. Even though the caffeic acid is in its oxidized form, the synthesized biomaterials possessed high antimicrobial activity and the SP6Ay samples were the ones that had an improved activity, especially against *E. coli*. Finally, the UV/Vis kinetic studies revealed that the drug is released with a two-step mechanism and the increase in P amount inside the systems led to a decrease in the release rate.

Even if all these results are promising, a deep investigation will be performed in order to evaluate the applicability of the above-synthesized SPxAy hybrid biomaterials on eucaryotic cells (e.g., osteoblast, fibroblast).

**Author Contributions:** Conceptualization, M.C.; methodology, A.D. and V.V.; software, L.V. and M.R.; validation, M.C., L.G.; formal analysis, L.V. and M.R.; investigation, A.D. and V.V.; data curation, M.C. and L.G.; writing—original draft preparation, M.C.; writing—review and editing, M.C.; visualization, M.C.; supervision, M.C. All authors have read and agreed to the published version of the manuscript.

**Funding:** This research received no external funding.

**Institutional Review Board Statement:** Not applicable.

**Informed Consent Statement:** Not applicable.

**Data Availability Statement:** The data presented in this study are available on request from the corresponding authors.

**Acknowledgments:** This work was supported in part by "SCAVENGE" financed by Università degli Studi della Campania Luigi Vanvitelli in the framework of "Piano Strategico di Ateneo 2021–2023—Azione strategica R1.S2".

**Conflicts of Interest:** The authors declare no conflict of interest.

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
